# Brownian dynamics simulation of protofilament relaxation during rapid freezing

**Evgeniy V. Ulyanov**[1], **Dmitrii S. Vinogradov**[2], **J. Richard McIntosh**[3], **Nikita B. Gudimchuk**[1,3,4]*

**1** Department of Physics, Lomonosov Moscow State University, Moscow, Russia, **2** Center for Theoretical Problems of Physicochemical Pharmacology, Moscow, Russia, **3** Department of MCD Biology, University of Colorado, Boulder, CO, United States of America, **4** Dmitry Rogachev National Medical Research Center of Pediatric Hematology, Oncology and Immunology, Moscow, Russia

* nikita_gb@mail.ru

**Data Availability Statement:** All relevant data are within the manuscript and its Supporting Information files

**Funding:** Russian Foundation for Basic Research grant # 20-34-70159 to N.B.G. https://www.rfbr.ru/

## Abstract

Electron cryo-microscopy (Cryo-EM) is a powerful method for visualizing biological objects with up to near-angstrom resolution. Instead of chemical fixation, the method relies on very rapid freezing to immobilize the sample. Under these conditions, crystalline ice does not have time to form and distort structure. For many practical applications, the rate of cooling is fast enough to consider sample immobilization instantaneous, but in some cases, a more rigorous analysis of structure relaxation during freezing could be essential. This difficult yet important problem has been significantly under-reported in the literature, despite spectacular recent developments in Cryo-EM. Here we use Brownian dynamics modeling to examine theoretically the possible effects of cryo-immobilization on the apparent shapes of biological polymers. The main focus of our study is on tubulin protofilaments. These structures are integral parts of microtubules, which in turn are key elements of the cellular skeleton, essential for intracellular transport, maintenance of cell shape, cell division and migration. We theoretically examine the extent of protofilament relaxation within the freezing time as a function of the cooling rate, the filament's flexural rigidity, and the effect of cooling on water's viscosity. Our modeling suggests that practically achievable cooling rates are not rapid enough to capture tubulin protofilaments in conformations that are incompletely relaxed, suggesting that structures seen by cryo-EM are good approximations to physiological shapes. This prediction is confirmed by our analysis of curvatures of tubulin protofilaments, using samples, prepared and visualized with a variety of methods. We find, however, that cryofixation may capture incompletely relaxed shapes of more flexible polymers, and it may affect Cryo-EM-based measurements of their persistence lengths. This analysis will be valuable for understanding of structures of different types of biopolymers, observed with Cryo-EM.

## Introduction

Over the recent years, there has been a burst of ground-breaking studies by cryo-EM of various intracellular structures, including microtubules–one of the three main filaments of the

rffi/eng Russian Federation Presidents' grant #-
1869.2020.4 from the Ministry of Science and
Higher Education of the Russian Federation to N.B.
G. http://government.ru/en/department/388/events/
Funders had no role in the study design, data
collection and analysis, decision to publish, or
preparation of the manuscript

**Competing interests:** The authors have declared
that no competing interests exist.

cytoskeleton [1–4]. Composed of 13 laterally attached tubulin protofilaments (Fig 1A), microtubules can exist in growing or shortening states, enabling them to search, capture and segregate chromosomes in mitosis [5]. The dynamic properties of microtubules are tightly coupled to the structure and chemical composition of the tubulin protofilaments at their tips [6]. It has been widely accepted that the tips of shortening microtubules contain curved protofilaments, but the structure of growing microtubule tips has remained controversial [7]. Recently, we and others have demonstrated that the ends of growing microtubules have a similar curved morphology, both in vitro and in cells from several species [8–13]. These observations have suggested a novel pathway of microtubule assembly. In addition, by analyzing images from electron cryotomography, we have discovered an unexpected feature of tubulin protofilaments. Protofilaments on both growing and shortening microtubules displayed a gradient of curvature: their tips are as highly curved as single tubulin dimers, but protofilament curvature becomes progressively less as one moves along the protofilament toward the microtubule lattice (Fig 1B–1F) [8].

The observed gradient in curvature on a single strand of protein molecules is surprising, since protein structure changes are usually very fast [14]. Microtubules consistently displayed protofilaments flaring out in planes that contained the microtubule axis and separated from their neighbors by ~27 degrees [8]. This angular separation is great enough to prevent lateral interactions between bending protofilaments, except very near the base of the protofilament (where it starts to flare) or indirectly through hydrodynamics. Lateral interactions are therefore unlikely to affect protofilament curvatures. In this context, we have sought explanations for the curvature gradient in aspects of sample preparation. One possibility, considered here in detail, is that the protofilaments, whose shapes must oscillate from thermal bombardment, may not have had time for complete relaxation to their minimum energy shape during rapid freezing. The rates of cooling during cryofixation are commonly $10^4$–$10^6$ K/s and can be even higher, depending on the method of freezing and sample thickness [15, 16]. The viscosity of supercooled water rises as the temperature drops [17]. It is therefore conceivable that the free tips of flexible filaments relax to their equilibrium shapes sooner than other parts of the filament, which should experience more viscous drag, thanks to their length.

In this manuscript, we model the relaxation of tubulin protofilaments from straight to curved shapes to see whether and how rapid freezing could affect their apparent curvatures. We also extend this analysis more broadly, by exploring potential effects of cryofixation on the apparent shapes and persistence lengths of other biological filaments.

## Materials and methods

Tubulin protofilaments were modeled as 2D chains of spheres (Fig 2A). Each sphere represented a tubulin monomer. α and β tubulins were treated identically. The length of the protofilament in the simulation was equal to 16 tubulin monomers, which is comparable to the average length of protofilament curls at the growing and shortening microtubule tips in vitro [8, 10]. One dimer at the minus-end of the protofilament was held in a constrained position to represent the microtubule lattice.

The monomers in the protofilament were bound to each other with two kinds of harmonic energy potentials. The first depended on the angle θ between each pair of adjacent tubulin monomers (Fig 2B):

$$W(\theta) = \frac{b}{2}(\theta - \theta_0)^2, \tag{1}$$

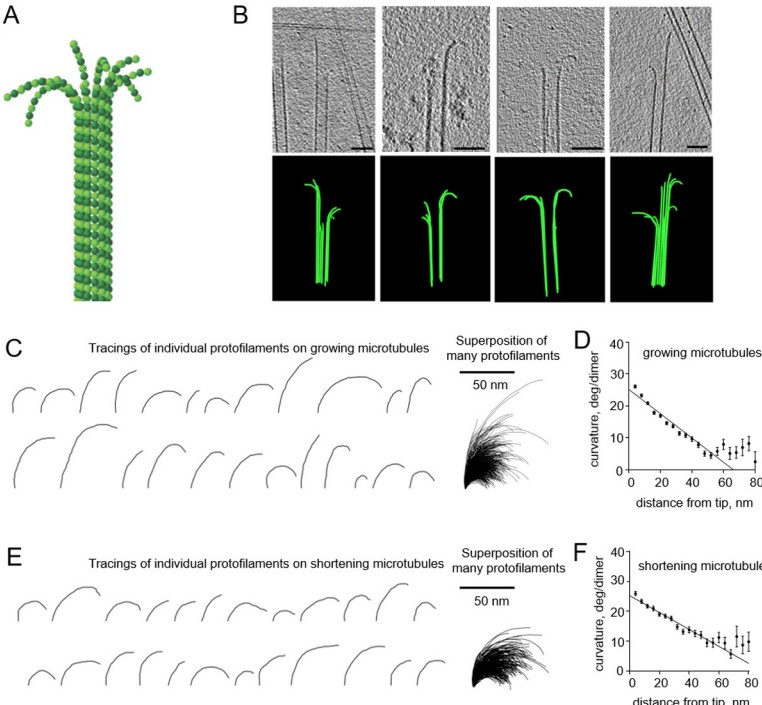

**Fig 1. Curvature gradient in tubulin protofilaments, observed by electron cryo-tomography.** All experimental data are from McIntosh et al., 2018 [8]. (A) Schematic of a microtubule, consisting of 13 αβ-tubulin protofilaments, flared at their tips. α-tubulins and β-tubulins are shown as darker and lighter green balls, respectively. (B) Four examples of growing microtubule ends, examined by electron cryo-tomography. The upper row of images displays tomographic slices of microtubule ends; the lower row shows corresponding models of reconstructed microtubules. Scale bars are 50 nm. (C) Representative examples of tubulin protofilament traces, based on electron cryo-tomographic analysis of microtubule ends, assembled in presence of 20 μM tubulin. The rightmost image shows a collection of 884 protofilaments that were analyzed in that experimental condition. (D) Dependence of protofilament curvature at the ends of growing microtubules on the distance from the protofilament tip. (E) Gallery of representative examples of tubulin protofilament traces, based on electron cryo-tomographic analysis of disassembling microtubule ends. The rightmost image shows a collection of the 335 protofilaments that were analyzed in this experimental condition. (F) Dependence of protofilaments curvature at the shortening microtubule ends on the distance from their tips.

where $W(\theta)$ is the energy of tubulin rotational strain, $b$ is the flexural rigidity coefficient, $\theta$ - the rotation angle relative to the adjacent subunit below, $\theta_0$ –equilibrium curvature.

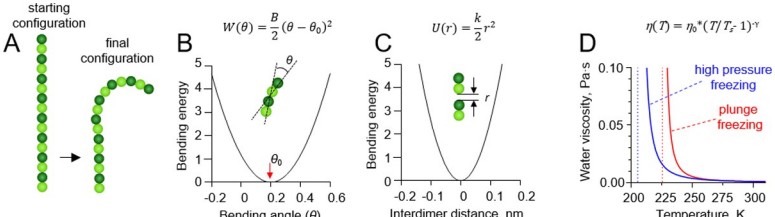

**Fig 2. Details of protofilament freezing simulations.** (A) Brownian dynamics model of a tubulin protofilament, consisting of seven movable dimers with one additional dimer in the bottom, whose position was fixed. The starting configuration of the protofilament is straight, the fully relaxed configuration is bent. The final configuration is usually bent, but the extent of bending depends on the rate of cooling and protofilament flexural rigidity. (B) Dependence of the inter-monomer bending energy on the angle between adjacent tubulin monomers. (C) Dependence of the inter-dimer interaction energy on the distance between interaction sites on the longitudinal interface of two tubulin dimers. (D) Red and blue lines show the dependences of water's viscosity coefficient on temperature in the plunge freezing (at atmospheric pressure) and high pressure freezing simulations (200 MPa), respectively. Dashed lines correspond to apparent water solidification temperatures in those types of simulations ($T_s$).

The second energy potential was keeping the monomers in the protofilament together (Fig 2C):

$$U(r) = \frac{k}{2} r^2, \qquad (2)$$

where $r$ is the distance between the longitudinal interaction points on the tubulin spheres, $U$ is the energy of longitudinal bond stretch, $k$ is the harmonic stiffness coefficient for inter-tubulin bonds.

Motions of the tubulin spheres were simulated using a Brownian dynamics approach [18]. Specifically, the two coordinates of the centers of mass of each tubulin sphere $(x,y)$ and the angular coordinate $(\theta)$, describing the orientation each sphere, were propagated forward in time, according to the following set of equations [19]:

$$x_i = x_{i-1} - \frac{dt}{6\pi R\eta(T)} \cdot \frac{\partial U_{total}}{\partial x} + \sqrt{2k_B T \frac{dt}{6\pi R\eta(T)}} \cdot N(0,1), \qquad (3)$$

$$y_i = y_{i-1} - \frac{dt}{6\pi R\eta(T)} \cdot \frac{\partial U_{total}}{\partial y} + \sqrt{2k_B T \frac{dt}{6\pi R\eta(T)}} \cdot N(0,1), \qquad (4)$$

$$\theta_i = \theta_{i-1} - \frac{dt}{8\pi R^3\eta(T)} \cdot \frac{\partial U_{total}}{\partial y} + \sqrt{2k_B T \frac{dt}{8\pi R^3\eta(T)}} \cdot N(0,1). \qquad (5)$$

Subscripts below the coordinates indicate the simulation iteration number, $dt$ is the time step of the simulation; $R = 2$ nm is the radius of the tubulin sphere, $\eta(T)$ is the temperature-dependent water viscosity coefficient; $T$ is temperature; $k_B$–is the Boltzmann constant; $N(0,1)$ is a normally distributed random number.

The temperature, $T$, in the simulations decreased over time, while the water viscosity coefficient, $\eta$, simultaneously increased. Following recent measurements [17], we implemented following a power-law dependence of viscosity on temperature:

$$\eta(T) = \eta_0 * (T/T_s - 1)^{-\gamma}, \qquad (6)$$

where $\eta_0$, $T_s$, $\gamma$ are some empirically determined parameters. In simulations of the plunge-freezing process, which is carried out at atmospheric pressure, we used the values, found by Dehaoui and colleagues [17]: $\eta_0 = 1.38 \cdot 10^{-4}$ Pa·s; $T_s = 225.6$ K; $\gamma = 1.64$. To simulate freezing at high pressure, we modified the parameters of the curve to provide the best fit to experimental measurements of water viscosity dependence on temperature at 200 MPa [20]: $\eta_0 = 2.2 \cdot 10^{-4}$ Pa·s; $T_s = 205.4$ K; $\gamma = 1.84$.

Practically, the steep increase of water viscosity with temperature resulted in complete immobilization at a temperature of about ~ -45 °C, well above the temperature of the liquid ethane in plunge freeze experiments (-165 °C) or liquid nitrogen jets (-196 °C) during high pressure freezing. When the sample enters the liquid ethane in drop freezing experiment or when the sample is cooled with a jet of liquid nitrogen, it is moving fast enough relative to the cryogen that it is continuously exposed to fresh cryogen at this temperature. Therefore, while the sample is cooling from 22 °C to ~ -45 °C (still much warmer than the cryogen), we considered the cooling rate, $v$, to be approximately constant. Given this assumption, the temperature

was decreased linearly in all simulations:

$$T = T_0 - vt, \tag{7}$$

where $T_0$ is the starting temperature, $v$ is the rate of cooling, $t$ is elapsed simulation time.

To assess the effects of cryofixation on the measurements of persistence length of soft polymers, we simulated Brownian motions of 100 filaments, each composed of 16 spherical subunits of 2.4 nm diameter, and with zero intrinsic curvature. In this case, the flexural rigidity, $b$, was equal to 3.3 kcal mol$^{-1}$ rad$^{-2}$. The initial configurations of the filaments were straight. Persistence lengths were measured as described previously [8], after 0.1 s of simulations at 310 K or after complete immobilization of the filaments as a result of either simulated high-pressure freezing at $10^4$ K/s, or simulated plunge freezing at either $10^6$ K/s or $10^7$ K/s.

Simulations were performed using custom C++ code, available upon request. Data were analyzed in Matlab 2017b.

All experimental data, presented in this manuscript, were obtained previously [8]. Detailed protocols of the ultrastructural analysis of microtubule ends are published in [21].

# Results

## Causing curvature gradients in tubulin protofilaments by cryofixation would require extremely high cooling rates

To gain insight into possible effects of cryofixation on the configurations that tubulin protofilaments might adopt, we carried out Brownian dynamics simulations of the relaxation of these protofilaments from a straight to curved configurations. This kind of relaxation is expected to occur when a protofilament is peeling from the straight microtubule lattice, which we believe happens with high frequency during both the assembly and disassembly of microtubules, thanks to thermal perturbations of protofilament shape.

During our simulations, the temperature was gradually decreased; as it approached the water solidification point, the model water viscosity coefficient increased, eventually leading to a complete immobilization of the protofilament. The dependence of the water viscosity coefficient on temperature was implemented with a power law dependence, as described in recent experimental studies [17, 20] (Fig 2D).

More rigid protofilaments are expected to achieve their relaxed shapes faster than softer ones. Experimental estimates of a tubulin protofilament's persistence length, a convenient measure of its flexural rigidity, span from 0.2 to 4.2 μm [8, 22, 23]. We picked two well-separated values from this range to represent the behavior of hypothetical 'floppy' and 'rigid' tubulin protofilaments. The 'floppy' protofilaments had the flexural stiffness of 58 kT/rad$^2$, corresponding to persistence length about 0.5 μm; the 'rigid' one had the flexural stiffness of 300 kT/rad$^2$, corresponding to about 2.5 μm of persistence length. These values were also used in our previous theoretical investigations of microtubule dynamics [8, 10, 24].

First, we modeled cryofixation at a moderate cooling rate, $10^4$ K/s, which is reportedly achieved during high pressure freezing of cellular samples [16] (S1 Video). In contrast to other liquids, the viscosity of water does not increase with pressure, but rather decreases by about up to one-half under 200 MPa pressure [20]. Therefore, high pressure is not expected to accelerate the cryo-immobilization process. Analysis of the protofilament shapes upon immobilization under such conditions revealed no dependence of their curvatures on the distance from the protofilament tip (Fig 3).

We then modeled cryofixation of protofilaments at a higher cooling rate, $10^6$ K/s, which can supposedly be achieved during plunge freezing of relatively thin samples [15] (S2 and S3

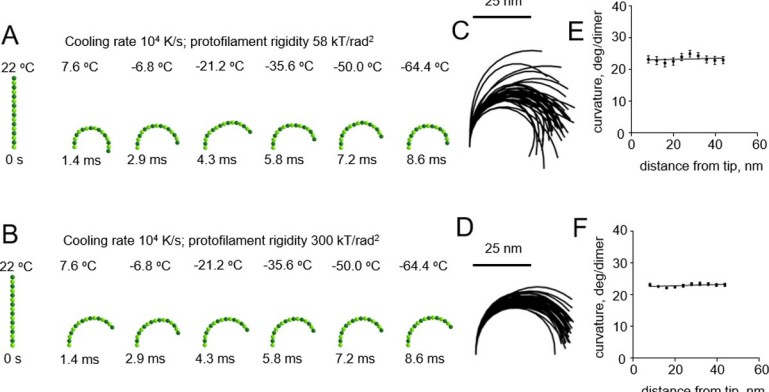

**Fig 3. Modeling protofilament cryofixation by high-pressure freezing (cooling rate 10⁴ K/s).** (A) Consecutive snapshots from a simulation of a 'floppy' protofilament as it freezes. Numbers above the protofilaments indicate the current temperature, numbers below show elapsed time (B) Same as panel A, but with a 'rigid' protofilament. (C) and (D) are ensembles of protofilament traces, based on simulations with the 'floppy' and 'rigid' protofilaments, respectively. Data are mean ± s.e.m. based on 40 independent simulations in each case. (E) and (F) are dependences of the protofilament curvature on the distance from the free tip, based on 40 simulations with the 'floppy' and the 'rigid' protofilament, respectively.

Videos). This too revealed no dependence of protofilament curvature on the distance from the protofilament tip (Fig 4).

A pronounced effect of cryofixation on the filament curvature became apparent only when we used an even higher cooling rate of $10^7$ K/s, which had been estimated to be theoretically possible only with extremely thin samples [16] (Fig 5A and 5B and S4 and S5 Videos). As expected, the gradient of curvature was higher with the floppy protofilaments than with the more rigid ones (Fig 5C–5F). In the range of plausible protofilament lengths, between 56 and 120 nm, the freezing-induced curvature gradients depended only weakly on the length of the protofilament (S1 Fig).

Based on this theoretical analysis, and given existing experimental estimates of PF flexural rigidity [8, 23, 25, 26], we conclude that for practical cases, such as high pressure freezing with

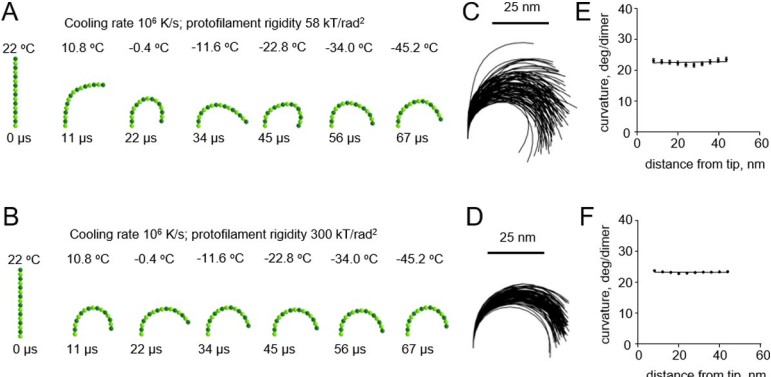

**Fig 4. Modeling protofilament cryofixation by plunge freezing (cooling rate 10⁶ K/s).** (A) Consecutive snapshots from a simulation of 'floppy' protofilament freezing. Numbers above the protofilaments indicate the current temperature, numbers below show elapsed time (B) Same as panel a, but with a 'rigid' protofilament. (C) and (D) are ensembles of protofilament traces, based on simulations with the 'floppy' and 'rigid' protofilaments, respectively. Data are mean ± s.e.m. based on 100 independent simulations in each case. (E) and (F) are dependences of the protofilament curvature on the distance from the free tip, based on 100 simulations with the 'floppy' and the 'rigid' protofilament, respectively.

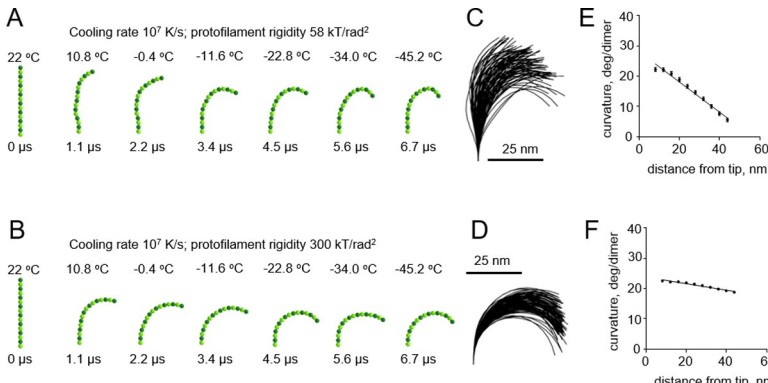

**Fig 5. Modeling protofilament freezing at an ultrahigh cooling rate ($10^7$ K/s).** (A) Consecutive snapshots from a simulation of a 'floppy' protofilament freezing. Numbers above the protofilaments indicate current temperature, numbers below show elapsed time. (B) Same as panel A, but with a 'rigid' protofilament. (C) and (D) are ensembles of protofilament traces, based on simulations with the 'floppy' and 'rigid' protofilaments, respectively. Data are mean ± s. e.m. based on 100 independent simulations in each case. (E) and (F) are dependences of the protofilament curvature on the distance from the free tip, based on 100 simulations with the 'floppy' and the 'rigid' protofilament, respectively.

$10^4$ K/s or plunge freezing with $10^6$ K/s, the cryofixation process is too slow to generate the tubulin protofilament curvature gradient.

## Analysis of tubulin protofilament shapes when samples are fixed under different experimental conditions

Our modeling suggests that tubulin protofilament curvature gradients, which we have found both in vivo and vitro [8], are not likely to be caused by cryofixation. To probe this issue more deeply, we returned to previously collected data and processed them to quantitatively compare protofilament curvatures on growing microtubule tips in *S. pombe* cells that were prepared at different freezing rates: by plunge or high pressure freezing (Fig 6A). The steepness of the observed curvature gradients were almost identical in those two freezing conditions. Importantly, a curvature gradient of comparable steepness was also observed when microtubules growing in vitro were prepared by chemical fixation (Fig 6B). These observations support our modeling conclusion that tubulin protofilament curvature gradients are unlikely to originate as an artifact of rapid freezing.

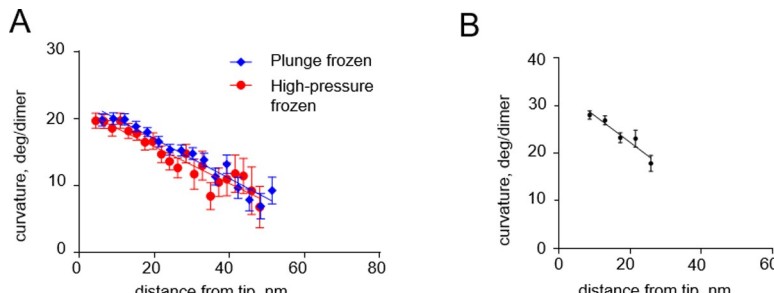

**Fig 6. Curvature gradients in tubulin protofilaments at the tips of growing microtubules, immobilized by high-pressure freezing and chemical fixation.** (A) Dependence of the protofilament curvature on the distance from the free tip, at the growing microtubule ends in high-pressure frozen or plunge frozen *S. pombe* cells. (B) Dependence of the protofilament curvature on the distance from the free tip, measured at growing microtubule ends in vitro, after chemical fixation with glutaraldehyde followed by imaging with negative stain electron tomography. All experimental data are from McIntosh et al. [8].

## Softer filaments can be captured in their incompletely relaxed states with lower cooling rates

Tubulin protofilaments are fairly rigid compared to other biological filaments, e.g. coiled coils with about 150 nm persistence length [27], DNA with persistence length about 45 nm [28], or collagen with persistence length of 10–25, depending on its type [29]. To generalize our analysis and facilitate its potential application to other polymer structures, we carried out additional simulations with filaments, whose persistence lengths ranged from ~66 nm to about 2.1 μm. As with tubulin protofilaments, we assumed that the equilibrium curvature of the softer polymers was non-zero, but the initial configuration was straight. This type of simulation corresponds to the situation, in which the cooling process itself, or some external factors, can trigger a conformational change in the polymer, launching its relaxation to a new equilibrium shape. As expected, in these simulations, softer protofilaments required lower cooling rates to induce a curvature gradient that is quantitatively similar to the one observed with more rigid filaments at higher cooling rates (Fig 7A). This suggests that incomplete relaxation could be captured by cryofixation of flexible filaments even at the cooling rates achievable in relatively thick biological samples [16].

## Predicting cryofixation effects on the apparent persistence length of polymers

If a polymer is not experiencing a dramatic conformational change during freezing, unlike tubulin protofilaments, our modeling approach can be used to estimate the expected effects of cryofixation on its apparent flexibility. Polymer flexibility is often characterized with a persistence length, $P$, which is simply related to its bending stiffness, $B$, through equation:

$$P = B/k_B T, \qquad (8)$$

where $k_B$ is the Boltzmann constant and $T$ is absolute temperature. Thus, even if $B$ is constant, persistence length is predicted to increase when temperature decreases. Therefore, if the relaxation of the filament to the equilibrium shape is faster than the cryofixation process, the persistence length of the cryo-immobilized polymer should appear larger than it really is. In order to illustrate this effect, we simulated freezing of a polymer with the persistence length of 20 nm, which is similar to that collagen. The simulation was carried out in two steps. First, we put the ~40 nm-long polymers into perfectly straight shape, the we let them fully relax to their equilibrium configurations at room temperature for 0.1 second. Next, we simulated high-pressure freezing of the polymer at $10^4$ K/s, drop freezing at either $10^6$ K/s or $10^7$ K/s. The apparent

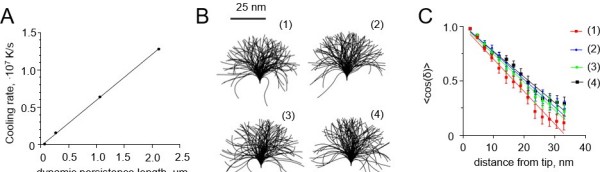

**Fig 7. Dependence of the freezing-induced curvature gradients and apparent persistence length on the cooling rate.** (A) Dependence of the cooling rate that is required to generate a curvature gradient of ~0.03 deg·nm$^{-2}$ on the flexural rigidity of a filament. Each point is based on 20 independent simulations. (B) Shapes of ensembles of 100 flexible polymers: (1) at 310 K before freezing; (2) after high-pressure freezing at $10^4$ K/s; (3) after freezing at $10^6$ K/s; (4) after freezing at $10^7$ K/s. (C) Dependence of the mean cosine of the angle of deflection of the filament from the straight shape on the distance from the filament tip. The data were used to determine persistence lengths of the filaments from panel B. Colors correspond to different freezing simulation parameters, as explained in panel B.

persistence length of the polymer increased to 28 ± 2 nm, 29 ± 1 nm, and 26 ± 2 nm, respectively (Fig 7B and 7C). These results suggest that under all three freezing conditions, the polymers had enough time to adopt configurations, characteristic for the temperature near water solidification point. Therefore, at least in the case of short polymers, freezing could affect the apparent flexibility of the polymers in cryo-EM-based measurements.

## Discussion

Samples prepared for cryoEM could exhibit cryofixation artifacts that are difficult to identify. Since most students of biological structure have almost complete faith in the validity of rapid freezing for sample preparation, an analysis that tests the possibility of structure change during freezing may have significant value. We have therefore used Brownian dynamics modeling to describe possible effects of cryofixation on the curvature and apparent flexibility of biological filaments. Here we have focused on microtubule protofilaments as test structures, whose physical properties can be modeled with some confidence, allowing us to evaluate the fidelity of the images obtained with this methodology. However, we have also applied our theory to other filaments, which are softer than tubulin protofilaments, and which do not necessarily possess intrinsic curvature.

In the case of microtubules, because of the high flexural rigidity of tubulin protofilaments, our analysis predicts that high pressure and plunge freezing are unlikely to induce the curvature gradients that are observed; these cooling rates are not fast enough to capture incompletely relaxed protofilament conformations. Only cooling rates as fast as ~$10^7$ K/s or higher, which are barely achievable in experiments, could, in theory, contribute to tubulin protofilament curvature gradient formation. This is consistent with our analysis of tubulin protofilaments in the experimental samples, prepared by different fixation methods. These gradients in protofilaments curvature are found not only in rapidly frozen samples viewed by cryo-EM but also in samples prepared by other methods [8]. Thus, they are probably a real feature of tubulin protofilament structure.

However, our analysis suggests that cryofixation could lead to two kinds of artifacts in observations of softer filaments. The first is the formation of curvature gradients in soft polymers, which may be relevant if the early stages of the cooling process itself, or some external factors, can trigger a conformational change in the polymer, launching its relaxation to a new equilibrium. If the cooling is fast enough, then the relaxation of the entire filament will not be complete before solidification of the specimen. As a result, the observed polymer configuration will not correspond to an equilibrium shape. For example, we predict that for polymers having persistence lengths comparable to DNA or collagen, curvature gradients should be observed with freezing at $10^4$–$10^6$ K/s. Among the possible reasons for triggering a conformational change by cooling is the effect of temperature on non-covalent bonds, such as those depending on the hydrophobic effect, or properties of the buffer, such as its dielectric constant [30, 31]. As previously discussed by Bednar and colleagues, this change could alter the electrostatic repulsion between DNA segments and thus the effective thickness of this polymer [28]. As a result, the state of molecules in cryo-vitrified specimens could differ from that at ambient temperature. In agreement with this idea, it had been reported that the helical repeat of short linear DNA fragments had a lower number of base-pairs per turn upon cryofixation [32]. Quantitatively, the extent of that structural effect was comparable to the effect of decreasing temperature from 20 to 10° C, suggesting that the DNA molecules had enough time to react to the rapid cooling even though cooling was expected to be very rapid in the 50 nm thick samples examined.

A second possible artifact of cryofixation, predicted by our model, is an increase in the apparent persistence length of polymers, observed with cryo-EM. Our modeling results

indicate that at least with short polymers, this kind of increase is expected even when the polymers are frozen at the fastest achievable cooling rates. It is noteworthy that the bending flexibility of the polymer (*B* in Eq 7) could additionally depend on temperature, as is the case for DNA [33]. This kind of effects is not considered by our theory. Another potentially important but currently unaccounted factor, which might affect filament relaxation during cryofixation, is hydrodynamic interactions [34].

Finally, besides pointing to potential artefacts of rapid freezing, our analysis raises the interesting possibility of creating probes to measure the rates of cooling during cryofixation, using relatively flexible polymers, whose intrinsic curvature could be rapidly switched at the onset of cryofixation by some external condition. Such conditions might include a rapid environmental change (addition of salt, change of pH), as was recently shown for collagen [35]; or irradiation with light, which has been used to create submicrometer bent pillar structures from photoreconfigurable azopolymer [36]. The extent of the cooling-rate-dependent relaxation of these polymers during cryofixation could then be calibrated to report site-specific freezing rates in the biological samples.

## Supporting information

**S1 Fig. Independence of the freezing-induced curvature gradient on the length of the filament.** Data for 56, 88, and 120 nm-long filaments with persistence length about 2.1 μm, whose freezing was simulated with the cooling rate of $1.28 \cdot 10^7$ K/s.
(TIF)

**S1 Video. Simulation of the relaxation of a 'floppy' tubulin protofilament during high-pressure freezing with $10^4$ K/s cooling rate.** Flexural rigidity is 58 kT/rad$^2$.
(MP4)

**S2 Video. Simulation of the relaxation of a 'floppy' tubulin protofilament during rapid freezing with $10^6$ K/s cooling rate.** Flexural rigidity is 58 kT/rad$^2$.
(MP4)

**S3 Video. Simulation of the relaxation of a 'rigid' tubulin protofilament during rapid freezing with $10^6$ K/s cooling rate.** Flexural rigidity is 300 kT/rad$^2$.
(MP4)

**S4 Video. Simulation of the relaxation of a 'floppy' tubulin protofilament during rapid freezing with $10^7$ K/s cooling rate.** Flexural rigidity is 58 kT/rad$^2$.
(MP4)

**S5 Video. Simulation of the relaxation of a 'rigid' tubulin protofilament during rapid freezing with $10^7$ K/s cooling rate.** Flexural rigidity is 300 kT/rad$^2$.
(MP4)

## Acknowledgments

Simulations were carried out using the equipment of the shared research facilities of HPC computing resources at Lomonosov Moscow State University.

## Author Contributions

**Conceptualization:** J. Richard McIntosh, Nikita B. Gudimchuk.

**Funding acquisition:** Nikita B. Gudimchuk.

**Investigation:** Evgeniy V. Ulyanov, Dmitrii S. Vinogradov, Nikita B. Gudimchuk.

**Methodology:** Nikita B. Gudimchuk.

**Software:** Evgeniy V. Ulyanov.

**Supervision:** Nikita B. Gudimchuk.

**Visualization:** Evgeniy V. Ulyanov.

**Writing – original draft:** Nikita B. Gudimchuk.

**Writing – review & editing:** J. Richard McIntosh, Nikita B. Gudimchuk.

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
