## [Decision Letter · Decision Letter 0]

20 Aug 2020

PONE-D-20-22086

Brownian dynamics simulation of tubulin protofilament relaxation during rapid freezing

PLOS ONE

Dear Dr. Gudimchuk,

Thank you for submitting your manuscript to PLOS ONE. After careful consideration, we feel that it has merit but does not fully meet PLOS ONE’s publication criteria as it currently stands. Therefore, we invite you to submit a revised version of the manuscript that addresses the points raised during the review process.

We look forward to receiving your revised manuscript.

Kind regards,

Jinhui Tao, Ph.D.

Academic Editor

PLOS ONE

Journal Requirements:

Reviewers' comments:

Reviewer's Responses to Questions

**Comments to the Author**

1. Is the manuscript technically sound, and do the data support the conclusions?

Reviewer #1: Partly

Reviewer #2: Yes

Reviewer #3: Yes

2. Has the statistical analysis been performed appropriately and rigorously? 

Reviewer #1: Yes

Reviewer #2: Yes

Reviewer #3: Yes

3. Have the authors made all data underlying the findings in their manuscript fully available?

Reviewer #1: Yes

Reviewer #2: Yes

Reviewer #3: Yes

4. Is the manuscript presented in an intelligible fashion and written in standard English?

Reviewer #1: Yes

Reviewer #2: Yes

Reviewer #3: Yes

5. Review Comments to the Author

Reviewer #1: In this work, the authors attempted to understand the observed microtubule structures after rapid freezing with Brownian dynamics simulation. it is interesting and meaningful for the communities to evaluate the potential artifacts of Cryo-EM based measurements in (bio)polymers.

However, as what the authors said in the discussion, 'tubulin protofilament curvature gradients are unlikely to originate solely as an artifact of rapid freezing.' Hence the manuscript has not convince me the general meanings and importance of the manuscript, and if the simulation is applicable to different kinds of (bio)polymers. The results do not agree with the experimental observation.

Besides, the authors discussed a lots of the perspective using the case of DNA, which is based on single data point of Fig. 5b. I am doubt if the the supportive is solid or if there are any experimental evidences.

In summary, I think the work is interesting. But the impact of the results is weak or has not been presented efficiently.

Reviewer #2: It is an interesting study to use Brownian dynamics modeling to investigate possible cryo-immobilization effects on the apparent curvatures of tubulin protofilaments from straight to curved shapes. It is found that the cooling rate and the flexural rigidity can influence the curvature gradients. This work provides useful information for the cryo-TEM samples preparation and shape analysis. I suggest that this paper can be accepted after a minor revision by addressing following questions:

1. In Figure 1b, a scale bar should be provided for cryo-TEM images.

2. In Figure 1d, f, why do the curvature far from tip close to 5-10 deg/dimer. Following my understanding, it should close to zero.

3. From Figure 3c-f, the authors suppose that cooling rate (106K/s) was not enough to produce a curvature gradient in protofilament tip, and higher cooling rate (106K/s) could lead to the formation of gradient (Figure 4c-f). But they concluded that “tubulin protofilament curvature gradients are unlikely to originate solely as an artifact of rapid freezing” from Figure 6. So how do we distinguish which reason results in the formation of curvature gradient when analyze them? Could the authors provide more explanations?

4. In the model, a single filament was used to model the curvature change, but the authors failed to consider how the interactions between multiple protofilaments affect the curvature gradient, as shown in Figure 1a and b.

5. The equations in Manuscript should provide references to support.

Reviewer #3: In this study, Ulyanov et. al. used Brownian dynamics modeling to theoretically examine possible cryo-immobilization effects on the apparent shapes of tubulin protofilaments, the key elements of the cellular skeleton. After their theoretically examined the extent of protofilament relaxation within the freezing time, they analyzed the microtubule curvatures and flexibilities, they conclude, that tubulin protofilament curvature gradients are unlikely to originate solely as an artifact of rapid freezing. The study is interesting, however, this referee has following comments co,

Major comments,

1. Line 105; The molecules are 3D objects. Thus the energy between two adjacent tubulin monomers should depended on their 3D angles, in which, other than the bending angle, the tilting angle (perpendicular to the bending angle) and twisting angle (rotation along the tilting axis) should be also considered.

2. Line 106: How B was defined in the simulation

3. Line 115: How k was defined in the simulation

4. Line 138: The temperature decreasing was assumed as the linear function during the cryofixation. It is not accurate assumption. In the early state of cryofixation, the sample surrounded ethane could be evaporated and the ethane gas will surround the sample/grid and reduce the sample cooling speed until the sample temperature is dropped below -88C, the liquified temperature of ethane. Author should discuss how the unified temperature influence on the flexibility of measurement.

5. Line 214: NS tomography study of DNA-nanogold showed the persistence length of DNA portion is ~ 116 to ~160 (depended on different types of bending angle measurement), while the all atom MD simulation confirmed the persistence length of ~150. (Nat Comm, 2016), which 2-3 times high than different from that measured from the cryo-EM 2D projection (~45, in your ref 28, Line 214), and SAXS measurement (~50, ref: Biophys. J. 97, 1408–1417 (2009). Authors should compare the variety with their simulation result as an evaluation or validation of the method.

6. PLOS authors have the option to publish the peer review history of their article (what does this mean?). If published, this will include your full peer review and any attached files.

Reviewer #1: No

Reviewer #2: No

Reviewer #3: No

---

## [Author Response · Author response to Decision Letter 0]

8 Dec 2020

We appreciate the careful reading of the manuscript and critical comments provided by the reviewers. Based on this criticism we have substantially re-written the manuscript and added several new simulations. Briefly, we have re-written the abstract, introduction and discussion sections to clarify the general logic of the paper. We have the carried out additional calculations, leading to new Figure 3, Figures 7B,C, and video S1. Please, note that because of these additions, the order of figures is changed. Former Fig. 5A is now transferred to Supplementary Fig.1. The new figure numbers are the ones cited here. Significant new text additions that address specific reviewer’s criticism are highlighted in red in the marked-up version of the manuscript. Small text improvements were not tracked to avoid clutter. Below, please, find our detailed, point-by-point response to each of the reviewers’ comments in blue. Video legends are provided in the marked-up copy of the Supplementary information. We think that these changes have significantly improved the manuscript and added value of the paper, and we hope that the reviewers agree.

Reviewer #1: In this work, the authors attempted to understand the observed microtubule structures after rapid freezing with Brownian dynamics simulation. it is interesting and meaningful for the communities to evaluate the potential artifacts of Cryo-EM based measurements in (bio)polymers.

However, as what the authors said in the discussion, 'tubulin protofilament curvature gradients are unlikely to originate solely as an artifact of rapid freezing.' Hence the manuscript has not convince me the general meanings and importance of the manuscript, and if the simulation is applicable to different kinds of (bio)polymers. The results do not agree with the experimental observation.

Response: 

We have substantially re-written the manuscript and added new simulations of high-pressure freezing (new Fig. 3) to make the logic of the paper more clear. In fact, the simulation results do not contradict experimental observations. Rather, our simulations predict that cryfixation process, happening at the rates, achievable during high pressure or plunge freezing, should not artificially induce any noticeable curvature gradients in rigid polymers like tubulin protofilaments. Therefore, the tubulin curvature gradients, which we previously described in McIntosh et al JCB 2018, must have another origin. As we hope now is now stated clearly in the paper, the gradients are very likely to be a real structural feature of tubulin protofilaments and may play some physiological role. Our modeling prediction is supported by our new comparative analysis of our previously collected data (Fig. 6). In full agreement with the model, it reveals that the tubulin protofilament curvature gradients in different experimental conditions are almost identical, regardless of the cooling rate and fixation method. This confirms the model conclusions that the freezing conditions commonly used for cryofixation are too slow to generate tubulin curvature gradients. The model, however, does predict that freezing might affect softer filaments, like DNA or collagen. 

Reviewer 1: Besides, the authors discussed a lot of the perspective using the case of DNA, which is based on single data point of Fig. 5b. I am doubt if the supportive is solid or if there are any experimental evidences. 

Response: 

The linear relationships between cooling rate and the dynamic persistence length, presented in former Fig 5b (new Fig 7b), are based on 4 points. Each point is based on analysis of 20 independent simulation repeats. 

To strengthen our point about the potential value of our Brownian dynamics approach to analysis of cryofixation effects on other polymers, we have carried out additional simulations (new Fig 7B,C), demonstrating apparent change of persistence length of soft polymers, frozen with a set of cooling rates. This suggests that cryofixation effects are not negligible and they should be taken into account in cryo-EM-based measurements of the flexibility of biological polymers. 

Reviewer #2: It is an interesting study to use Brownian dynamics modeling to investigate possible cryo-immobilization effects on the apparent curvatures of tubulin protofilaments from straight to curved shapes. It is found that the cooling rate and the flexural rigidity can influence the curvature gradients. This work provides useful information for the cryo-TEM samples preparation and shape analysis. I suggest that this paper can be accepted after a minor revision by addressing following questions:

1. In Figure 1b, a scale bar should be provided for cryo-TEM images.

Response: 

Fixed. Thank you.

Reviewer #2: 

2. In Figure 1d, f, why do the curvature far from tip close to 5-10 deg/dimer. Following my understanding, it should close to zero.

Response: 

The reviewer is correct. In the ideal case, one should expect zero, but we traced only the parts of PFs, which were visually 'curved', hence the lowest “observed” curvature is non-zero. 

Reviewer #2:

3. From Figure 3c-f, the authors suppose that cooling rate (106K/s) was not enough to produce a curvature gradient in protofilament tip, and higher cooling rate (106K/s) could lead to the formation of gradient (Figure 4c-f). But they concluded that “tubulin protofilament curvature gradients are unlikely to originate solely as an artifact of rapid freezing” from Figure 6. So how do we distinguish which reason results in the formation of curvature gradient when analyze them? Could the authors provide more explanations?

Response:

We have substantially re-written the manuscript and added new simulations of high-pressure freezing (Fig 3, Video S1) to make the logic of the paper more clear. Please, also see our response to a similar point of Reviewer 1.

Reviewer #2:

4. In the model, a single filament was used to model the curvature change, but the authors failed to consider how the interactions between multiple protofilaments affect the curvature gradient, as shown in Figure 1a and b.

Response:

This is a very good point. However, our observations of microtubule structure consistently showed protofilaments flaring out in planes that contained the microtubule axis and separated from their neighbors by ~27o (Mcintosh et al., 2018). This angular separation is great enough to prevent lateral interactions between bending protofilaments, except very near the base of the PF (where it starts to flare) or through hydrodynamics. If some aspect of our observations is faulty (failing to reveal PFs that stick to one another over some fraction of their length), then PF interaction might be a factor. We have added a note about this to the introduction.

Reviewer #2:

5. The equations in Manuscript should provide references to support.

Response:

We have added more references and to explain and support the equations better.

Reviewer #3: In this study, Ulyanov et. al. used Brownian dynamics modeling to theoretically examine possible cryo-immobilization effects on the apparent shapes of tubulin protofilaments, the key elements of the cellular skeleton. After their theoretically examined the extent of protofilament relaxation within the freezing time, they analyzed the microtubule curvatures and flexibilities, they conclude, that tubulin protofilament curvature gradients are unlikely to originate solely as an artifact of rapid freezing. The study is interesting, however, this referee has following comments 

Major comments,

1. Line 105; The molecules are 3D objects. Thus the energy between two adjacent tubulin monomers should depended on their 3D angles, in which, other than the bending angle, the tilting angle (perpendicular to the bending angle) and twisting angle (rotation along the tilting axis) should be also considered.

Response:

Here we considered a 2D case, which was essentially dictated by planar shape of the protofilament curls in cryo-ET [McIntosh et al JCB 2018, Gudimchuk et al., Nat Commun. 2020]. Moreover, in many cases, cryo-EM produces 2D projections of polymers, so our 2D analysis will be applicable. We believe that a full account of 3D would not affect the outcome of the analysis, but it would add some new unknown parameters. 

Reviewer #3:

2. Line 106: How B was defined in the simulation

Response:

We explained the meaning of B, and varied it in the range of plausible values. 

Reviewer #3:

3. Line 115: How k was defined in the simulation

Response:

Thank you. We have added a definition of k. 

Reviewer #3:

4. Line 138: The temperature decreasing was assumed as the linear function during the cryofixation. It is not accurate assumption. In the early state of cryofixation, the sample surrounded ethane could be evaporated and the ethane gas will surround the sample/grid and reduce the sample cooling speed until the sample temperature is dropped below -88C, the liquified temperature of ethane. Author should discuss how the unified temperature influence on the flexibility of measurement.

Response:

We state in the text that the linear decrease of temperature is an approximation, which allows using a notion of some 'constant cooling rate' (lines 154-161):

 "Practically, the steep increase of water viscosity with temperature resulted in complete immobilization at a temperature of about ~ -45 oC, well above the temperature of the liquid ethane in plunge freeze experiments (-165 oC) or liquid nitrogen jets (-196 oC) during high pressure freezing. When the sample enters the liquid ethane in drop freezing experiment or when the sample is cooled with a jet of liquid nitrogen, it is moving fast enough relative to the cryogen that it is continuously exposed to fresh cryogen at this temperature. Therefore, while the sample is cooling from 22 oC to ~ -45 oC (still much warmer than the cryogen), we considered the cooling rate, v, to be approximately constant."

This point is additionally illustrated in Fig. R1 below. These plots show that the linear approximation of the exponential curve works fairly well in the range of temperatures between room temperature and the temperature of the solidification (R2 =0.9914). 

Figure R1 (attached in the PDF file: response to reviewers). Justification of the approximation of the cooling process with a linear function 

Reviewer #3: 

5. Line 214: NS tomography study of DNA-nanogold showed the persistence length of DNA portion is ~ 116 to ~160 (depended on different types of bending angle measurement), while the all atom MD simulation confirmed the persistence length of ~150. (Nat Comm, 2016), which 2-3 times high than different from that measured from the cryo-EM 2D projection (~45, in your ref 28, Line 214), and SAXS measurement (~50, ref: Biophys. J. 97, 1408–1417 (2009). Authors should compare the variety with their simulation result as an evaluation or validation of the method.

Response:

Reviewer seems to be confusing the persistence length with another characteristic, ‘the bending energy’, used to describe DNA flexibility in the paper by Zhang et al., Nat. Comm. 2016 (DOI: 10.1038/ncomms11083). The values cited (~ 116 to ~160, ~150 and ~50) correspond to the estimates of the bending energies of DNA, expressed in kcal/mol in the paper, rather than the persistence length of DNA. In fact, Zhang et al explicitly stated that “the widely used persistence length of 50nm was used to compute the bending energy”. This value of persistence length is similar to what we cited in ref 28.

---

## [Decision Letter · Decision Letter 1]

1 Feb 2021

Brownian dynamics simulation of protofilament relaxation during rapid freezing

PONE-D-20-22086R1

Dear Dr. Gudimchuk,

We’re pleased to inform you that your manuscript has been judged scientifically suitable for publication and will be formally accepted for publication once it meets all outstanding technical requirements.

Kind regards,

Satoshi Honda

Academic Editor

PLOS ONE

Additional Editor Comments (optional):

Reviewers' comments:

Reviewer's Responses to Questions

**Comments to the Author**

1. If the authors have adequately addressed your comments raised in a previous round of review and you feel that this manuscript is now acceptable for publication, you may indicate that here to bypass the “Comments to the Author” section, enter your conflict of interest statement in the “Confidential to Editor” section, and submit your "Accept" recommendation.

Reviewer #1: All comments have been addressed

Reviewer #2: All comments have been addressed

Reviewer #3: All comments have been addressed

2. Is the manuscript technically sound, and do the data support the conclusions?

Reviewer #1: Yes

Reviewer #2: Yes

Reviewer #3: Partly

3. Has the statistical analysis been performed appropriately and rigorously? 

Reviewer #1: (No Response)

Reviewer #2: Yes

Reviewer #3: Yes

4. Have the authors made all data underlying the findings in their manuscript fully available?

Reviewer #1: (No Response)

Reviewer #2: Yes

Reviewer #3: Yes

5. Is the manuscript presented in an intelligible fashion and written in standard English?

Reviewer #1: (No Response)

Reviewer #2: Yes

Reviewer #3: Yes

6. Review Comments to the Author

Reviewer #1: The manuscript has been improved a lot.

I only suggest the authors furhter polishing the written English.

Reviewer #2: (No Response)

Reviewer #3: In the revised manuscript, authors have addressed some of this referee’s concerns. However, considering the simulation result is challenging one of very important experimental conclusions in the field of cryo-EM, following comments are needing to be addressed in the manuscript.

Major comments,

1. Authors’ stated the limitation of their method, which should be included in the manuscript, such as “Here we considered a 2D case, which was essentially dictated by planar shape of the protofilament curls in cryo-ET [McIntosh et al JCB 2018, Gudimchuk et al., Nat Commun. 2020]. Moreover, in many cases, cryo-EM produces 2D projections of polymers, so our 2D analysis will be applicable. We believe that a full account of 3D would not affect the outcome of the analysis, but it would add some new unknown parameters.?”

2. Simulation did not include the influence to result from the phase transition of the buffer/water (from liquid to solid). Is it unclear at what temperature that liquid turn to solid? After the liquid turn to solid, the free movement of sample will be no longer allowed in the simulation. The ignored the influence from the solid state of buffer may cause a serious problem in the conclusion. Authors should include the discussion of the limitation of phase transition to the method in the manuscript.

3. Moreover, authors should include two important control simulations, the relaxing of the conformation/flexibility under a stable room temperature, and under a stable temperature that author believed this temperature is the phase transition temperature of the buffer/water. This control simulation is important to evidence that the curvature is not due to the relaxation process of the simulation

4. It is important to authors to include some discussion about whether the PFs flare showed in the negative-staining, AFM, and heavy metal shadowing experiments, which did not involve in any temperature changing during sample preparation.

7. PLOS authors have the option to publish the peer review history of their article (what does this mean?). If published, this will include your full peer review and any attached files.

Reviewer #1: No

Reviewer #2: No

Reviewer #3: No

---

## [Editor Report · Acceptance letter]

3 Feb 2021

PONE-D-20-22086R1 

Brownian dynamics simulation of protofilament relaxation during rapid freezing 

Dear Dr. Gudimchuk:

I'm pleased to inform you that your manuscript has been deemed suitable for publication in PLOS ONE. Congratulations! Your manuscript is now with our production department. 

Kind regards, 

on behalf of

Dr. Satoshi Honda 

Academic Editor

PLOS ONE